# A New TDR-Based Sensing Cable for Improving Performance of Bridge Scour Monitoring

**DOI:** 10.3390/s20226665

**Published:** 2020-11-21

**Authors:** Kai Wang, Chih-Ping Lin, Wei-Hao Jheng

**Affiliations:** 1Department of Civil Engineering, National Chiao Tung University, No. 1001 Ta-Hsueh Rd., Hsinchu City 300, Taiwan; kaiwang@nctu.edu.tw; 2Yijhu Township Office, No. 389, Renli Village, Yijhu Town, Chiayi County 624, Taiwan; Jheng@mail.cyhg.gov.tw

**Keywords:** bridge scour, scour monitoring, time-domain reflectometry (TDR), TDR sensing waveguide

## Abstract

The use of time domain reflectometry (TDR) for real-time monitoring of bridge scour process has gone through several stages of development. The recently-proposed concept of bundled TDR sensing cable, in which two sets of insulated steel strands are twisted around and connected to a central coaxial cable to form a compact sensing cable, is a major change that has several advantages including the bottom-up sensing mechanism. Nevertheless, there is big room for improving its measurement sensitivity and signal to noise ratio (SNR). Changes in waveguide configuration also need to be made to avoid the adverse effect of insulation abrasion observed in field implementation. This study evaluated three new conductor and insulator configurations for constructing the sensing waveguide, including a balanced two-conductor waveguide (Type I), an unbalanced three-conductor waveguide with insulation coating on the middle conductor (Type II) and an unbalanced three-conductor with insulation coating on the two outer conductors (Type III). In all cases, the spacing between the two sets of steel strands (i.e., the waveguide conductors) was especially enlarged by replacing some steel strands with non-conductor wires to increases measurement sensitivity and avoid shorted conditions due to insulation abrasion. Experimental results show that Type III has the best performance on all counts. A new improved TDR sensing cable was hence proposed based on Type III configuration. Its performance was further evaluated by a full-scale experiment to take into consideration the long range of measurement in most field conditions. Detailed discussions on improvements of measurement sensitivity and SNR, limitation of sensing range, and mitigating the adverse effect of insulation abrasion are presented.

## 1. Introduction

Local scour is regarded as the top threat to bridge safety. More than half of bridge failures in the United States can be attributed to local scour [1,2,3]. In Taiwan, severe and rapid scour during typhoon seasons is the major concern of Taiwan’s bridge safety. Because of the lack of an effective real-time scour monitoring system, the bridge authority can only rely on water level thresholds to contingently close bridges as a precautionary measure during severe storms. Real-time monitoring of bridge scour is extremely useful for more realistically assessing bridge safety, as well as better understanding the scour mechanism, verifying the scour formula, and planning hydraulic countermeasures. Unfortunately, a bridge scour monitoring technique suitable for working in a harsh environment such as Taiwan’s severe flow and bed conditions is yet to be developed. Several instruments and techniques have been proposed to monitor bridge scour. Table 1 compares a list of existing bridge scour monitoring methods. Depending on the measurement principle, each type of method has its own advantages and limitations under certain environmental conditions. More recently, Boujia et al. [4] combined optical frequency domain reflectometry (OFDR) and the Winkler model to evaluate the effect of scour by performing distributed strain measurements along a rod under static lateral loads. Maroni et al. [5] developed a scour monitoring system consisting of smart probes equipped with electromagnetic sensors for monitoring both scour and deposition process around bridge foundations. However, the installation and operation of such devices under Taiwan’s severe flow conditions are not suitable. Fitzgeraldet et al. [6] developed a cantilever-based piezoelectric energy harvesting device for vibration-based bridge scour detection. Although being fully non-intrusive, the vibration-based method only detects the change in the structural response due to a certain level of scour. It may recognize the presence of scouring when it is so critical to affecting the structural stability instead of providing direct and more complete information on the scour process.

Aiming to monitor the scour process in harsh flow conditions, the TDR (time domain reflectometry) method especially draws our attention in recent years because it is a guided wave technique without electronic elements under water. Although it is still an intrusive method, it is considered to have great potential to better sustain severe flooding conditions and monitor the scour and deposition process. A TDR device sends out an electromagnetic pulse through a leading coaxial cable to a sensing waveguide and record reflections from the sensing section, which contains information regarding the changing interface (scour depth). The first attempt to use TDR technique for the purpose of scour monitoring can be found in the work of Dowding and Pierce [11], in which the coaxial cable is modified as a sensing waveguide by attaching cross flanges at certain intervals. Scour will expose part of the embedded flange cable and the exposed flanges shear and shorten the cable that is monitored by pulsing TDR. Because this kind of sensing mechanism is only valid for maximum scour depth and the tip of sheared-off cable becomes not waterproof, other types of TDR sensing waveguides for bridge scour monitoring have been proposed, including sensing rod type [7,12,13,14] and sensing wire type [15,16]. Similar to a TDR soil moisture content probe, Yankielun and Zabilansky [12] developed a TDR scour sensing probe consisted of two parallel stainless-steel tubes, which is probably the first generation of TDR scour sensing waveguide that actually works in the field. Without coating on the conductor(s) of the sensing waveguide, the conductive environment in the sediments limits the sensing length due to signal attenuation. A few modifications to improve the TDR scour sensing probe have been proposed [7,13,14]. More recently, a bigger and innovative change in constructing a TDR scour sensing waveguide is the bundled TDR sensing cable proposed by Wang et al. [15], in which two sets of PVC-insulated steel strands are twisted around a central coaxial cable. The central coaxial cable guides the wave to the bottom end and then connect to the sensing waveguide formed by the pair of steel strands. This new invention was further investigated numerically and experimentally for working in general field conditions [16]. In an actual field implementation, the monitoring system successfully captured the scour process during a storm event but also revealed some practical issues that need further improvement [16]. The PVC coating used to separate two opposing electrodes and reduce conductive loss also reduces the sensitivity to scour change. There is a need for maximizing the measurement sensitivity by re-designing the sensing waveguide. Furthermore, the PVC coating is likely to be abraded by the bed load. In its current form, the insulation coating may be partially worn off and unable to completely insulate the two opposing sets of steel strands. How coating abrasion affects the measurement or how to minimize the adverse effect of coating abrasion also needs further investigation. 

The idea of using a TDR-based sensor has been proposed for more than 20 years. However, the usefulness of such techniques and many others has been limited, especially in harsh flow conditions where scour is most critical. Existing TDR scour sensors are rod type, which is unfavorable in high-velocity flow and has a very small measurement range. The recent concept of bundled TDR sensing cable is a big step of change. However, the sensing performance was greatly sacrificed in pursuit of durability and workability. This study proposed a new TDR scour sensor through various waveguide considerations and thorough experimental evaluation, in order to improve the measurement sensitivity and confront the adverse effect of coating abrasion and the limitation of measurement range. Three new configurations of waveguide conductors and insulators were proposed and tested to compare their SNRs and measurement sensitivities. Two of the more promising designs were further examined for the effect of coating abrasion. Based on the experimental evaluation, an improved bundled TDR sensing cable was then recommended for future use. A full-scale sandbox experiment was conducted to reveal the measurement range of the new bundled TDR sensing cable with and without coating abrasion. The full-scale experimental data was also used to validate a new data reduction scheme to avoid the adverse effect of coating abrasion.

## 2. Background: TDR Scour Monitoring and Bundled Sensing Cable

Time domain reflectometry is a versatile waveguide-based technique that has been adapted for various geotechnical measurements [17,18,19,20]. A TDR device basically consists of a step pulse generator for emitting electromagnetic (EM) waves into the transmission line, a sampler for receiving the reflected signals, and an optional oscilloscope for displaying instant waveforms. The transmission line can be divided into the leading coaxial cable and the sensing section made of a sensing waveguide (e.g., TDR scour sensing cable used in this study). For the purpose of bridge scour monitoring, Wang et al. [15] recently proposed a bundled TDR sensing cable, as shown in Figure 1a. The sensing cable was constructed by six steel strands with half of them insulated by PVC coating or jacket to form the opposing electrodes for the sensing waveguide. The bottom-up sensing is advantageous, and it is achieved by wrapping the sensing waveguide around a central coaxial cable. The incident wave from a TDR device is guided by the central coaxial cable to the cable end before connecting to the sensing section formed by the six steel strands for bottom-up sensing. The crucial scour depth is the first impedance mismatch position subject to change when sensing bottom up. Furthermore, bottom-up sensing can effectively overcome the fouling effect in the air and water sections, and all the complicated multiple reflections from water/air (w/a) interface and cable end become irrelevant since they occur after the main reflection from the sediment-water interface. Figure 1b illustrates how a TDR bridge scour monitoring system can be installed in the field. Similar to soil anchors, the TDR sensing cable can be installed in a borehole with a grouted base anchor and extended up and fixed on a bridge girder or slab. It can be installed near a bridge foundation for local scour monitoring or between two bridge foundations for monitoring contraction scour. The TDR signal recorded by the sampler is often presented in terms of the reflection coefficient (*ρ*) as a function of time, which can be written as:(1)$$\rho (t)=\frac{v(t)-v_{i}}{v_{i}}$$
where *v* is the recorded voltage and *v_i_* is the magnitude of the incident step pulse. Figure 1c illustrates the behavior of a reflected waveform from a TDR sensing cable. As the characteristic impedance of a transmission line is inversely proportional to the squared root of dielectric constant, a negative reflection occurs at the sediment/water (s/w) interface followed by a positive reflection at the water/air (w/a) interface and an end reflection (negative for shorted end and positive for open end) at the cable end. The round-trip arrival times of these reflections are marked as *t_s/w_*, *t_w/a_*, and *t_e_*, respectively. The time *t*_0_ marks the reflection from the connection between the coaxial cable and sensing waveguide. 

By measuring the round-trip travel time (∆*t*) in a medium section and knowing the medium section length (*L*), the pulse propagation velocity (*V*), which is related to the apparent dielectric constant (*K_a_*), can be determined by the following equation [17]: (2)$$V=\frac{c}{\sqrt{K_{a}}}=\frac{2L}{\Delta t}$$
where *V* is the propagation velocity, *c* is the speed of light (2.998 × 10^8^ m/s), *K_a_* represents the apparent dielectric constant (in the case of bundled TDR sensing cable, it is affected by the insulation coating and medium surrounding the sensing cable), *L* is the length of a section and ∆*t* is the round-trip travel time in that section. To form a sensing waveguide and avoid conductive attenuation, part of the steel strands is insulated by PVC coating. The TDR sensing waveguide has a certain sampling range in the transverse direction. Within the sensing range of the TDR scour sensing cable, the dielectric media include the PVC coating and the surrounding media. The apparent dielectric constant (*K_a_*) in Equation (2) would be the effective average dielectric constant in each sensing section (i.e., sediment, water, and air). Due to this insulation coating, the apparent dielectric constants in different media (i.e., sediment, water, and air) become less distinct, resulting in weaker reflections at s/w and w/a interfaces comparing to the reflection at *t_0_*. To facilitate waveform interpretation, Wang et al. [15] proposed a time-lapse differential waveform method for more precisely identifying the location of scour depth (i.e., s/w interface). As shown in Figure 1d, a selected reference waveform (e.g., the initial reading) is subtracted from subsequent measurements to yield time-lapse waveforms in terms of the difference in reflection coefficient (Δ*ρ*). The new sediment/water interface (*t_s/w_*) due to scour can be easily identified as the first break in the differential waveform. With the measured *t_s/w_* and calibrated propagation velocity of sediment (*V_s_*), the crucial scour depth (in terms of length of sediment section *L_s_*) can be easily computed as:(3)$$L_{s}=\frac{(t_{s/w}-t_{0})*V_{s}}{2}$$

It is noted that the dielectric permittivity of water is slightly affected by temperature. Hence, vs. is theoretically also affected by temperature. However, the temperature effect is negligible for the sediment section due to small temperature variation under riverbed unless in the frozen state, in which dielectric constant drops significantly from the liquid state to the solid state.

## 3. Improving the Bundled TDR Sensing Cable

The bundled TDR sensing cable possesses good workability and enables the advantageous bottom-up sensing mechanism. However, the measurement sensitivity is greatly sacrificed due to the close proximity of insulated steel strands and opposing steel strands. Field implementation revealed an adverse effect of coating abrasion and a higher noise level than in a laboratory setting. The measurement sensitivity should be improved to enhance the signal to noise ratio to make the TDR method more robust. The first generation of bundled TDR sensing cable is a balanced type (two-conductor transmission line), which is prone to unwanted noise in the field [21]. Symmetric multi-conductor probes have been used to improve signal quality for soil water content measurements. On the other hand, the sensitivity weighting in the space depends on the conductor diameter and the spacing between two opposing conductors [22,23]. Therefore, the measurement sensitivity of TDR scour sensing cable may be further improved if the size and configuration of the sensing waveguide are modified. In its current form, the two opposing sets of steel strands are separated only by the PVC coating. Shorted condition, where two sets of steel strands are in touch, may be caused by coating abrasion. Re-designing the sensing waveguide is to avoid this abrasion problem while enhancing the measurement sensitivity at the same time. 

### 3.1. Reconfiguring Conductors and Insulators for Better Measurement Sensitivity and SNR

To avoid the abrasion problem and increase measurement sensitivity (or SNR), the conductors and insulators should be reconfigured. In addition to the original design, Figure 2 show three new configurations for the sensing waveguide. As depicted in Figure 2a, the original sensing cable consists of a central coaxial cable (50 ohm CFD-240), three PVC coated steel strands as one conductor, and three steel strands as the opposing conductor. The first type of improvement, Type I in Figure 2b, replaces two coated steel strands with two non-conductive Polypropylene (PP) wires. In doing so, the two sets of conductors are farther separated. This can avoid the undesirable shorted condition by abrasion and improve measurement sensitivity by greater conductor spacing. Type I is still a balanced type (two-conductor) transmission line, which is prone to field noise. By further replacing the middle steel strand of the three un-coated steel strands, a symmetric three-conductor transmission line can be formed, as shown in Figure 2c,d. The difference between Figure 2c (Type II) and Figure 2d (Type III) is that the PVC coating is on the middle conductor (connected to the central conductor of the coaxial cable) for Type II and on the outer conductor for Type III. The symmetric transmission line should have a lower noise level, but its sensitivity weighting is more localized near the conductors (partially affected by insulation coating) than a two-conductor transmission line. Therefore, the noise level may be reduced, but reflection signal strength may also be reduced. The sensitivity weighting is higher around the middle conductor, avoiding insulation coating on the middle conductor in Type III may improve the sensitivity to the surrounding media and reduce the effect of coating abrasion. In evaluating the waveguide configuration, 6 mm diameter steel strand and 1 mm thick PVC coating were used. The coaxial cable, PVC-coated steel strand and PP wire are all 8 mm in diameter. 

The signal quality in terms of SNR for the original sensing cable and the three newly-proposed ones were experimentally evaluated by a laboratory sandbox experiment. The SNR for this study is first defined as: (4)$$SNR=\frac{A}{\sigma }$$
where *A* is the amplitude of reflection signal from the sediment/water interface and *σ* is the standard deviation of the noise residing on the waveform. All sensing waveguides shown in Figure 2 are affected by the insulation coating and PP wires. Since the dielectric constant of insulators is much lower than that of saturated sediment, a higher effective apparent dielectric constant (lower propagation velocity) sensed by the waveguide in the sediment section would indicate higher measurement sensitivity. Both SNR and apparent dielectric constant were examined to compare the performance of different sensing waveguides. 

The schematic and a photo showing the sandbox experiment are shown in Figure 3a. A TDR device (Sympuls Aachen—Line Impedance Analyzer TDR 3000) was linked to the developed sensing cable via a 1 m long 50-ohm coaxial cable to record the TDR reflection signals. The measured waveform was sampled with 10,000 points to cover all the required reflection signals in the test, and the sampling interval was 10 ps. The number of waveform stacking (averaging) was 4. With this setting, the corresponding window length is 15 m, and the spatial resolution is about 1.5 mm. The experiment to evaluate measurement sensitivity was carried out in a 0.2 m diameter acrylic cylinder, where different water levels and sediment thicknesses were controlled. Each of the tested sensing cables was individually placed in the center of the acrylic cylinder, and subsequently added poorly-graded quartz sands (particle size ranging from 0.1 to 1.0 mm) with a 10 cm increment until maximum 1.0 m sediment thickness. The water level was maintained at equal to or greater than the maximum sediment thickness. At each sediment thickness increment, the corresponding TDR reflection signal was recorded and analyzed for evaluating its SNR value and measurement sensitivity. 

As an example, Figure 3b shows the recorded raw waveforms of the Type III TDR sensing cable. The actual characteristic impedance of the 1 m leading coaxial cable (outside the TDR sensing steel cable) is slightly different from the coaxial cable inside the TDR sensing cable. Because of this impedance mismatch, there is a clear negative step reflection at their connection; and multiple reflections due to this mismatch occur in the later part of the waveform, making it more difficult to clearly identify the sediment/water interface (*t_s/w_*). Using the maximum sediment thickness (*L_s_* = 1.0 m) as the reference case, Figure 4 shows the time-lapse differential waveforms for the four different TDR sensing cables. In all cases, the arrivals of reflections from sediment/water interfaces (denoted as *t_s/w_*) can be clearly identified after taking the waveform difference with the reference waveform. After *t_s/w_* is determined, the apparent dielectric constant (*K_a_*) in the sediment section can be determined from the round-trip travel time (Δ*t* = *t_s/w_ − t_0_*, where *t_0_* represents the arrival time of reflection from the connection between central coaxial cable and the sensing waveguide) and sediment thickness by Equation (2). On the other hand, the standard deviation of noise residing on the waveform is determined as the standard deviation of the flat waveform in the coaxial cable before the reflection from the sensing section as marked as ① in Figure 4a, and the illustration of determining the reflection amplitude from the sediment/water interface (A) is given by the mark ② in Figure 4a. The SNR can then be calculated by Equation (4).

Table 2 summarizes the results of SNR and measurement sensitivity (in terms of travel time and effective apparent dielectric constant in the sediment section) for the four different TDR sensing cables. All measured travel times were normalized to the travel time per 1.0 m change of sediment thickness for direct comparison. Since the dielectric constant of saturated sediment is larger than insulation materials, better measurement sensitivity of a TDR scour sensing cable will result in a longer travel time (Δ*t*) and a higher *K_a_* value under the same amount of sediment thickness (*L_s_*). According to the results of travel time Δ*t* and *K_a_*, all three new sensing waveguides have significantly better measurement sensitivity over the original TDR sensing cable. This can be attributed to the increase of spacing between two opposing conductors. The close proximity of the two sets of conductors in the original design makes the sensing area concentrated in the narrow space largely occupied by the PVC coating. The sensitivity weights much more near the middle conductor of an unbalanced waveguide (symmetric multi-conductor transmission line) in a way similar to that near the central conductor of a coaxial transmission line. This is probably why Type III (the waveguide with uncoated middle conductor) has further higher measurement sensitivity over Types I and II. Although Types I and II have similar measurement sensitivity, Type II has lower SNR because the balanced type is more prone to electrical noise. The noise level is smaller for both unbalanced waveguides (Types II and III). Because of its higher measurement sensitivity, Type III stands out as having the best SNR and measurement sensitivity. Both the unbalanced waveguides (Types II and III) were chosen to further examine the effect of coating abrasion before deciding the best design for the new TDR sensing cable. 

### 3.2. Effect of Coating Abrasion

The two opposing conductors in the new TDR sensing waveguides are separated by PP wires. Conductor coating is now unnecessary to form a sensing waveguide, but it is used to avoid signal attenuation due to DC conductivity in saturated sediment and water. A previous field implementation has shown that the bedload and debris may abrade the insulation coating. In the new design where two conductors are separated by PP wires, the shorted condition due to coating abrasion can now be avoided. Nevertheless, how coating abrasion affects TDR measurements in the new waveguide design needs further investigation. 

A laboratory abrasion experiment was conducted to examine the abrasion effect on the two improved sensing cables (i.e., Types II and III). The testing setup is basically the same as the previous experiment in testing measurement sensitivity and SNR. The testing procedure is illustrated in Figure 5. The water level was kept at 1.2 m from the end of the TDR sensing cable. From Step (1) to (3), sediments (quartz sands) were gradually added into the tank with a 0.1 m increment until the sediment height reached 1.2 m. The results can be used to simulate the normal scour and deposition process. At Step (4), sediments were removed to reduce the sediment height to 0.6 m again, and then a 5 cm long PVC coating (actually a PVC jacket in this study) was cut off near the sediment/water interface to simulate field abrasion. The abraded TDR sensing cable was soaked in water for two days before to examine the effect of water seeping into the PVC jacket. Finally, sediments were gradually added back to the acrylic cylinder to simulate the deposition under the condition of coating abrasion.

The electric field energy of an unbalanced TDR probe is concentrated near the middle conductor [22]. This means that TDR response is more sensitive to the material surrounding the middle conductor. Therefore, the response of Type II sensing waveguide, in which the middle conductor is covered with insulation material, may be affected more by insulation abrasion. Figure 6a,b shows the measured waveforms before abrasion, right after abrasion and two days after abrasions for Types II and III, respectively. The corresponding differential waveforms taking the waveform before abrasion as the reference are shown in Figure 6c,d. As expected, the Type III sensing waveguide is less affected by the coating abrasion as the waveform change by coating abrasion is much smaller than for Type II. Since there is no distinct shift in travel time at the abrasion position (s/w interface) two days after the initial coating abrasion, neither Type II nor Type III was affected by water seeping into the PVC jacket. Figure 7 shows the effect of coating abrasion after sediment refilling, which mimics what often occurs in the field. The differential waveforms subtracted by the waveform of maximum sediment thickness are compared with the results without coating abrasion (i.e., from results between Steps (1) and (3)). The first breaks that represent the locations of s/w interface are more significantly affected by coating abrasion in Type II than in Type III. Since Type III is less affected by coating abrasion in addition to having better measurement sensitivity, it was later chosen as the new improved design of TDR sensing cable.

### 3.3. The New Improved TDR Sensing Cable

Based on the Type III configuration, a new improved TDR sensing cable was proposed for future field applications. It will have better measurement sensitivity, higher SNR, and insensitivity to the effect of coating abrasion. The diameter of steel strands is enlarged to increase the overall tensile strength of the bundled TDR sensing cable, but not too much to impede cable handling. Furthermore, the HDPE is used instead of PVC and PP to increase its resistance against abrasion. The prototype of the new TDR sensing cable is shown in Figure 8. The new TDR sensing cable consisted of one 13 mm diameter coaxial cable (50 ohm CNT-400), one 12 mm diameter steel strands, two 9 mm diameter steel strands covered by 1.5 mm thick HDPE coating, and three 10 mm diameter HDPE wires, as shown in Figure 8a. The HDPE wires properly separate the three conductors to increase measurement sensitivity and avoid the shorted condition from coating abrasion. To form an unbalanced waveguide, the uncoated steel strand is connected to the inner conductor of the coaxial cable while the two coated steel strands are connected to the outer conductor of the coaxial cable. The unbalanced configuration further enhances the SNR and minimizes the effect of coating abrasion. Figure 8b shows a photo of the assembled TDR sensing cable. The performance experiment described in Section 3.1 was repeated for the new TDR sensing cable. The SNR, travel time per 1.0 m sand, and *K_a_* of the new TDR sensing cable are 41.2, 2.18 × 10^−8^ s, and 10.69, respectively. All indicators show that the new TDR sensing cable is superior to all waveguides shown in Figure 2. The improved measurement sensitivity of the new TDR sensing cable over the smaller Type III waveguide may be attributed to the increased diameter of steel strands and HDPE wires (i.e., increased conductor diameter and spacing). 

## 4. Performance of the Improved TDR Sensing Cable in Full Scale

Under a long range of measurement in most field conditions, cable resistance and dielectric loss will attenuate the electromagnetic signal and make the reflections smoother as sensing distance increases. A full-scale scour simulation experiment showed that the maximum measurement range of the original TDR scour sensing cable is about 6 m [16]. The performance of the new TDR bundled scour sensing cable, including the effect of coating abrasion, was further evaluated by a similar full-scale experiment. The experiment was performed in a 10 cm × 10 cm in cross section and 10 m long hollow acrylic channel, as shown in Figure 9. There are plastic baffles every 0.25 m to keep the new TDR sensing cable at the center and control the sediment thickness in 0.5 m increments. A TDR device (TDR 3000) was connected to a 13 m long new sensing cable via a 1 m long leading coaxial cable to acquire TDR reflection signals. The experimental procedure is illustrated in Figure 10. The 13 m long improved TDR sensing cable was placed at the center of the 10 m long acrylic channel with a constant water depth (length of the water section in the testing channel) at 10.0 m (Figure 10a). The poorly-graded quartz sand (particle size ranging from 0.1 to 1.0 mm) was added into the channel in 0.5 m increments (Figure 10b) until sediment thickness reached 10 m (Figure 10c), simulating variation of riverbed. As the sediment thickness increased, the corresponding TDR waveforms were acquired. The same procedure was repeated for the same TDR sensing cable but with coating abrasion between 3.25 and 3.75 m from the cable end (Figure 10d,e). 

### 4.1. Accuracy and Sensing Range of Scour Measurement

The measured raw data of the full-scale scour simulation experiment for the intact TDR sensing cable are shown in Figure 11a. Using the case of *L_s_* = 10 m as the reference waveform, the differential waveforms in Figure 11b represents changes due to different scour conditions. Compared to the original TDR scour sensing cable [16], the strength and SNR of the reflection signal from s/w interface have significantly improved. However, similar to the original TDR sensing cable, the reflection signal attenuates and becomes unidentifiable as the sediment thickness (*L_s_*) exceed 6 m. While the new scour sensing cable significantly improves the measurement sensitivity and SNR within the measurement range, the maximum measurement range remains similar. As sediment thickness increases, the reflection signal from the s/w interface gets weaker regardless of waveguide configuration. Although the DC electrical conduction path between two sets of conductors has been blocked by the insulation coating, signal attenuation with increasing sensing range may be attributed to frequency-dependent conduction effect and/or radiation loss. Hence, reconfiguration of sensing waveguide is unable to extend the sensing range of a single TDR sensing cable. A separate longer TDR sensing cable is needed to extend the measurement range in the field.

According to Equation (3), the bottom-up scour sensing scheme requires only two parameters (i.e., propagation velocity in sediments vs. and arrival time *t_s/w_*) for determining the crucial scour depth [15]. The propagation velocity vs. can be calibrated by at least one manual determination of *L_s_*. Figure 12 shows the relationship between 2*L_s_* and round-trip travel time (*t_s/w_ − t_0_*) in sediments from the small-scale experiments (*L_s_* within 1 m) and full-scale experiments (*L_s_* within 6 m). The linear relationship of Equation (3) is quite stable regardless of the testing scale. The calibration results of vs. from small-scale experiments (Figure 12a) and large-scale experiments (Figure 12b) are quite close, 9.48 × 10^7^ and 9.37 × 10^7^ m/s, respectively. Figure 13 shows the estimated *L_s_* in the full-scale experiments using both calibrated velocity. The estimation based on full-scale calibration is slightly better than the small-scale calibration. However, both estimations are within 5% error, suggesting that propagation velocity vs. can be determined either using small-scale calibration in the laboratory or more precise full range calibration in the field.

### 4.2. Effect of Coating Abrasion and Countermeasure

The second phase of the full-scale experiment was to investigate the effect of coating abrasion under a long sensing range. A 50 cm coating abrasion was introduced between 3.25 and 3.75 m from the end of the sensing cable. This may occur during a large flooding event and the sediments above the coating abrasion represent refilled sediments after the flood recedes. Taking the same initial waveform (intact sensing cable at *L_s_* = 10 m) as the reference, Figure 14 compares the differential waveforms of the abraded sensing cable with that of the intact sensing cable at different *L_s_* conditions. When the s/w interface under measurement is below the coating abrasion (e.g., *L_s_* = 1.0, 2.0 and 3.0 m), the first break in the differential waveform is not affected by the coating abrasion. It represents a deeper scour depth below the coating abrasion. When the sediments are refilled above the coating abrasion, the differential waveform sees a slight drop (or more of a tilt) at the coating abrasion before the reflection drop at the refilled s/w interface. The effect of coating abrasion in the full-scale experiment is more pronounced than the results in Figure 7 mainly because the 50 cm abrasion is much longer than the 5 cm abrasion in the small-scale experiment. Although the new TDR scour sensing cable has been redesigned to minimize the effect of coating abrasion, the interpretation of measured waveforms becomes more complicated after severe coating abrasion and sediment refilling. Fortunately, the monitoring of maximum scour depth, which is most critical to bridge safety, is unaffected by the coating abrasion since it is always below the previous abrasion location.

The time-lapse differential method usually takes the initial waveform (before any further scouring) as the reference to get rid of intrinsic multiple reflections from the measurement system and facilitate identification of *t_s/w_*, as can be seen by comparing Figure 11b with Figure 11a. Any change from the reference waveform, including coating abrasion, is reflected in the differential waveform. It should be noted that the propagation velocity in the refilled sediments may be slightly different from that of the original backfilled sands. If sediment refilling is significant after the current maximum scouring, the actual refilled thickness can be checked by an actual field survey. The propagation velocity in the refilled sediments can then be calibrated. Furthermore, the reference waveform for the time-lapse differential method can be updated by the latest waveform after the field survey to remove the insulation abrasion effect. As an example, taking the waveform after the coating abrasion and sediments refilled to *L_s_* = 7.0 m as the new reference waveform, Figure 15a shows the differential waveforms with the updated reference. Besides the intrinsic multiple reflections, the effect of coating abrasion can also be removed by the updated differential method. The subsequent scouring in the refilled sediments can be accurately determined by the updated vs. as well, as shown in Figure 15b. It should be emphasized that the measurement of maximum scour depth is not affected by the insulation effect even without updating the reference waveform because of the bottom-up sensing approach. 

## 5. Conclusions

A bundled TDR sensing cable was recently proposed, significantly altering the idea of how a TDR scour sensing waveguide may be constructed for harsh flow conditions, in which none of the existing methods can work properly for monitoring the continuous scour dynamics. However, the sensing performance was greatly sacrificed in pursuit of durability and workability. This study proposed a new TDR scour sensor through various waveguide considerations and thorough experimental evaluation, in order to improve the measurement sensitivity and confront the adverse effect of coating abrasion and the limitation of measurement range. Three new conductor and insulator configurations for constructing the sensing waveguide were investigated, including a balanced two-conductor waveguide (Type I), an unbalanced three-conductor waveguide with insulation coating on the middle conductor (Type II), and an unbalanced three-conductor with insulation coating on the two outer conductors (Type III). In all cases, the spacing between the two sets of conductors was especially enlarged by replacing some steel strands with non-conductor wires. This enlarged separation of two opposing conductors increases measurement sensitivity and avoids shorted condition due to coating abrasion. The performance of the new designs, in terms of measurement sensitivity, SNR and ability to reduce the effect of coating abrasion, was experimentally evaluated and compared. The results show that Type III has the highest measurement sensitivity and SNR. It is also least affected by coating abrasion. A new improved TDR sensing cable was hence proposed based on Type III configuration. 

The performance of the improved TDR sensing cable was further evaluated by a full-scale experiment to take into consideration the long-range of measurement in most field conditions. Compared to the original design, the new TDR sensing cable significantly improves the measurement sensitivity and SNR, but the maximum scour measurement range remains similar. As sediment thickness increases, the reflection signal from the sediment/water interface gets weaker regardless of waveguide configuration. Although the DC electrical conduction path between two sets of conductors has been blocked by the insulation coating, signal attenuation with increasing sensing range persists and may be explained by frequency-dependent conduction effect and/or radiation loss. Therefore, a separate longer TDR sensing cable is needed to extend the measurement range in the field. Within the measurement range (about 6 m), the bed location associated with scouring can be accurately estimated within 5% error. 

The coating abrasion does not affect the monitoring of maximum scour depth, which is most critical to bridge safety. However, the measured waveforms after more extensive coating abrasion and sediment refilling become more complicated and the effect of coating abrasion is quite noticeable in the differential waveform. To better monitor the subsequent scouring in the refilled sediments and remove the effect of abrasion effect on future monitoring results, a new data reduction approach was proposed that involves updating the reference waveform and calibration of propagation velocity in refilled sediments. This new approach was successfully validated by the full-scale experiment. The much-improved measurement sensitivity and SNR of the new TDR sensing waveguide together with the new data reduction approach enable more robust automation of data interpretation. Field implementation will be carried out in the near future. 

## Figures and Tables

**Figure 1 sensors-20-06665-f001:**
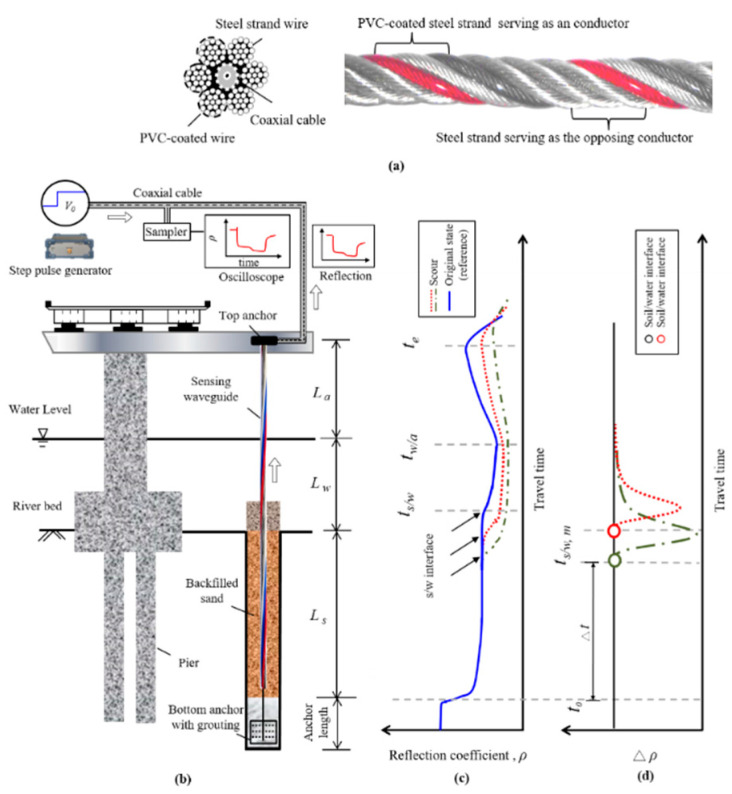
(**a**) Schematic and photo of the TDR sensing cable; (**b**) illustration of TDR bridge scour monitoring; (**c**) illustration of TDR bottom-up measurement waveforms and definition of various reflection arrival times; and (**d**) illustration of time-lapse differential waveform method to clearly identify reflection from the sediment/water interface.

**Figure 2 sensors-20-06665-f002:**
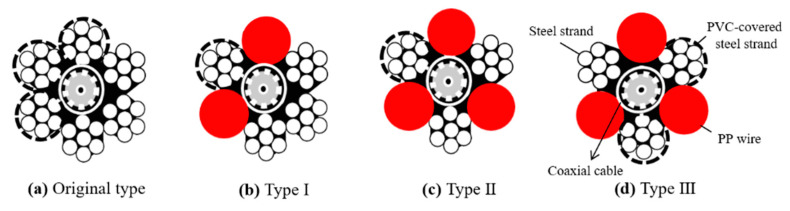
Waveguide configurations: the original TDR sensing cable (**a**); and the three newly-proposed types (**b**–**d**). All diameters are 8 mm except for the bare steel strand, which is 6 mm in diameter.

**Figure 3 sensors-20-06665-f003:**
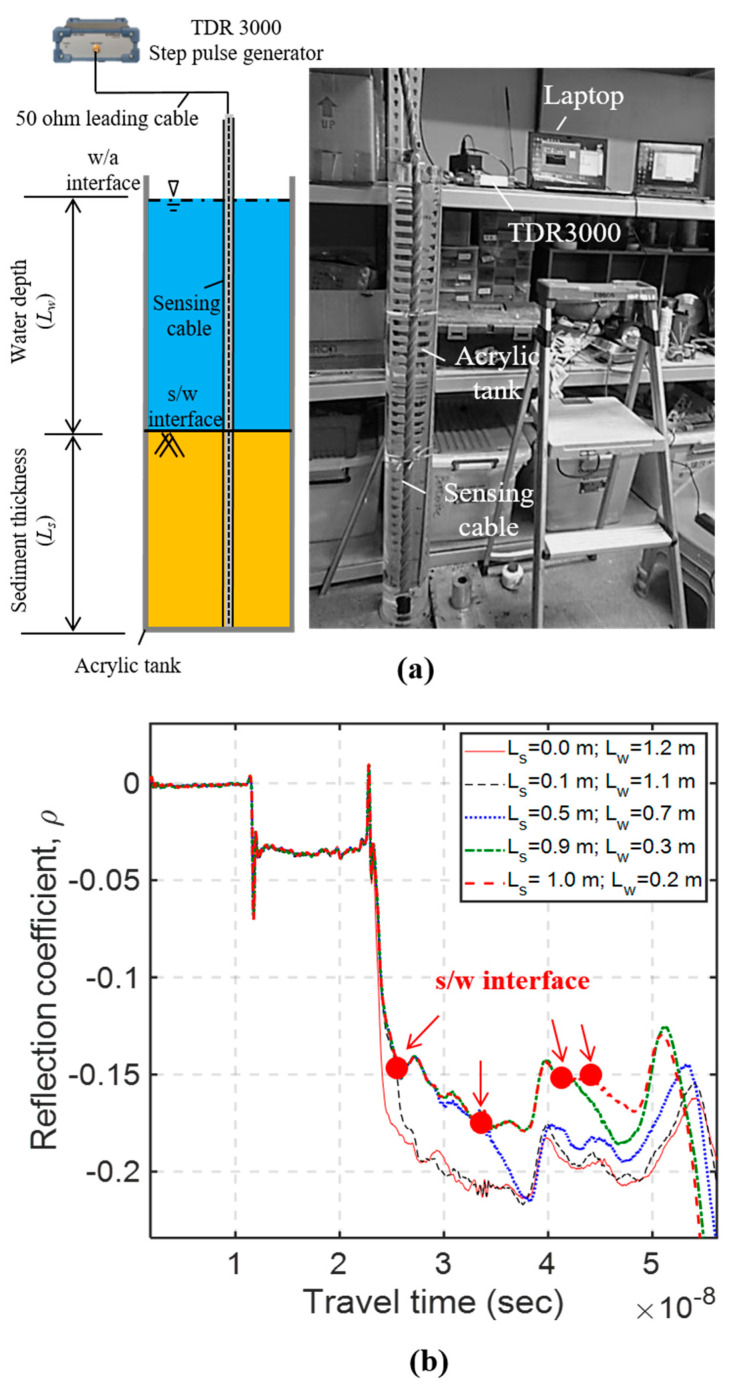
(**a**) Laboratory experiment setup; and (**b**) the recorded raw waveforms of the Type III TDR sensing cable.

**Figure 4 sensors-20-06665-f004:**
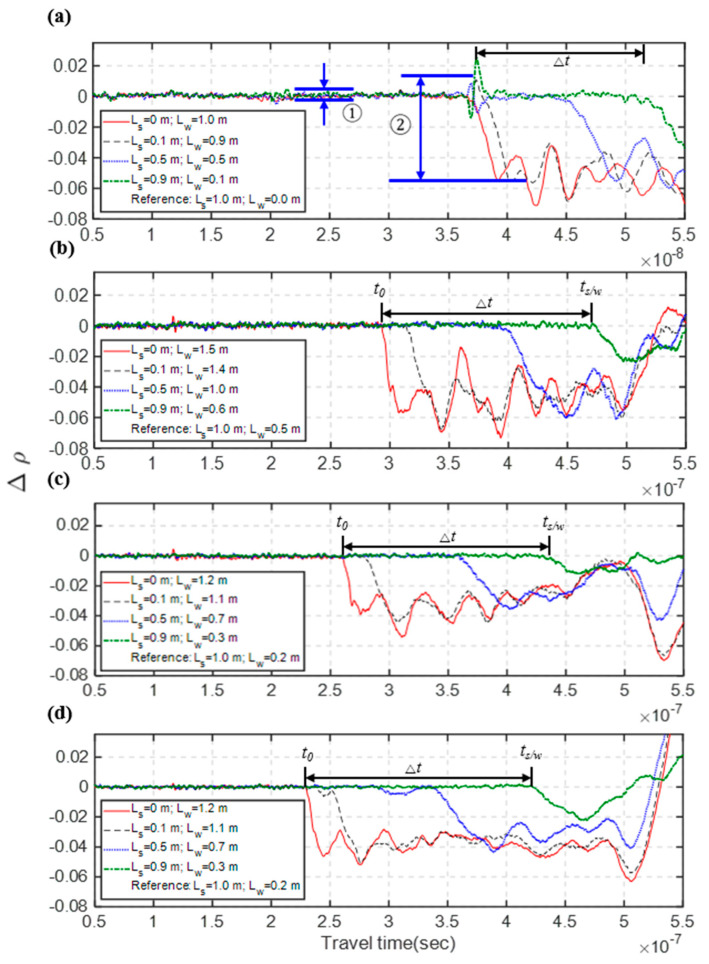
Results of differential waveform analysis in the scour sandbox experiment: (**a**) original TDR sensing cable with definitions of SNR and travel time in the sediment section; (**b**) Type I sensing waveguide; (**c**) Type II sensing waveguide; and (**d**) Type III sensing waveguide.

**Figure 5 sensors-20-06665-f005:**
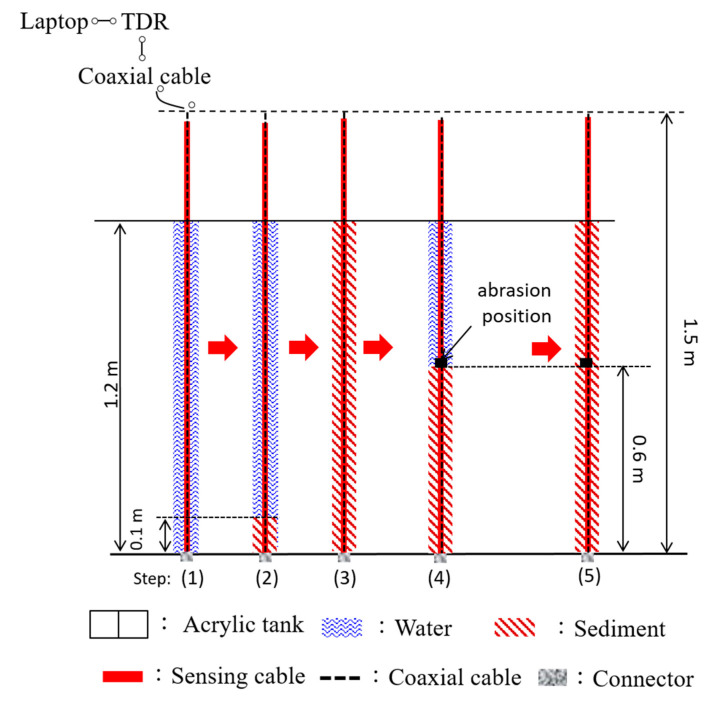
Illustration of the testing procedure to investigate the effect of coating abrasion.

**Figure 6 sensors-20-06665-f006:**
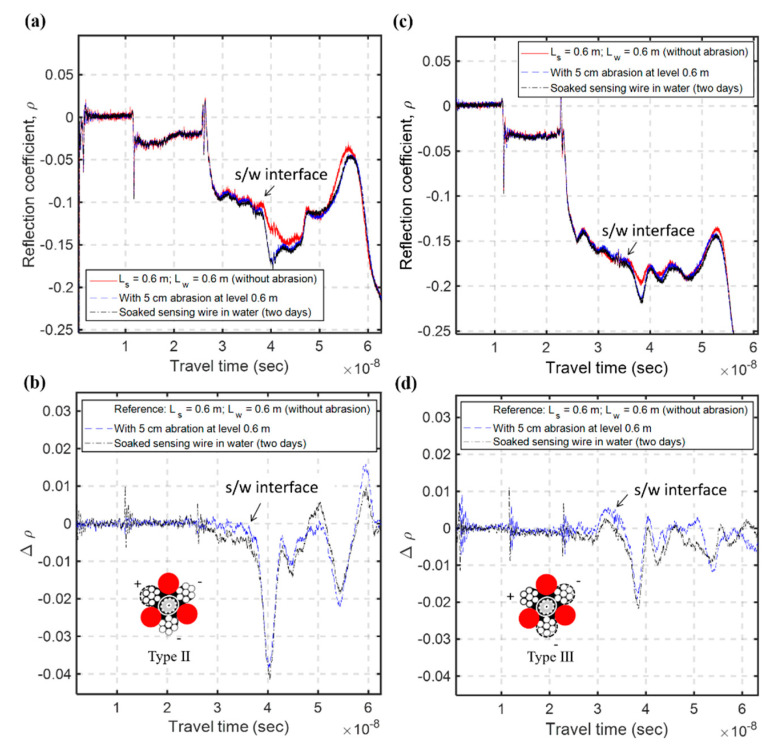
The measured waveforms before abrasion, right after abrasion, and two days after abrasions for Type II (**a**) and Type III (**b**) waveguides; and the corresponding differential waveforms taking the waveform before abrasion as the reference for Type II (**c**) and Type III (**d**) waveguides.

**Figure 7 sensors-20-06665-f007:**
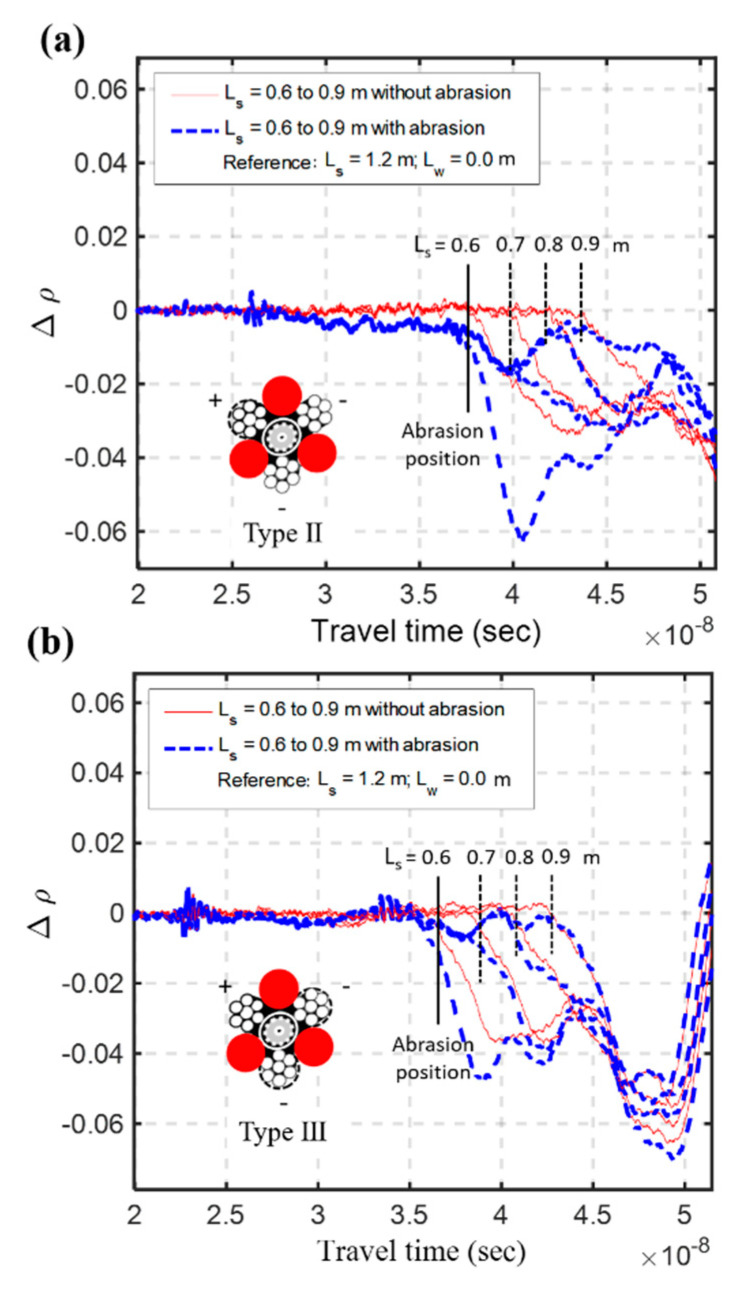
The effect of coating abrasion after sediment refilling for Type II waveguide (**a**) and Type III waveguide (**b**).

**Figure 8 sensors-20-06665-f008:**
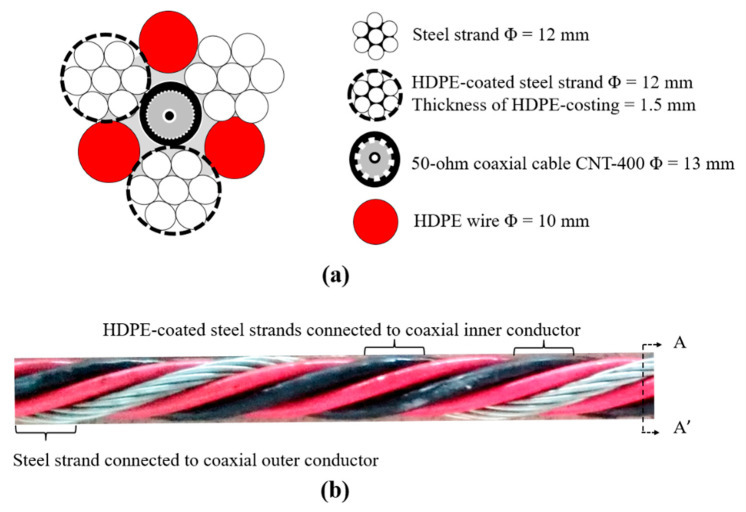
(**a**) The cross section, A-A’ of (**b**), showing the new waveguide configuration; and (**b**) a photo of the assembled new TDR scour sensing cable.

**Figure 9 sensors-20-06665-f009:**
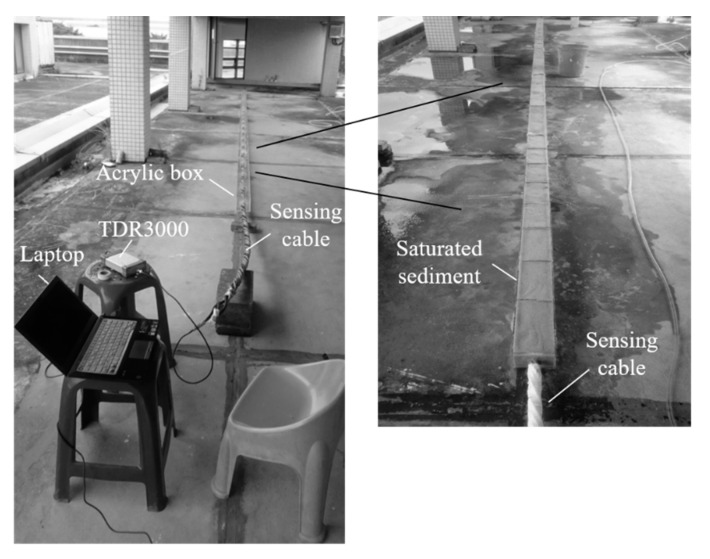
Photos showing the experimental setup of the full-scale scour simulation test.

**Figure 10 sensors-20-06665-f010:**
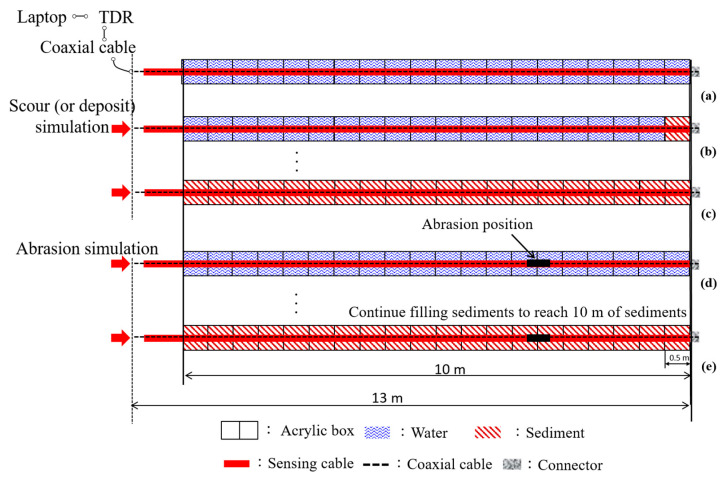
Illustration of the testing procedure of the full-scale scour simulation experiment: (**a**–**c**) testing sequence for the intact TDR sensing cable; and (**d**,**e**) testing sequence for the TDR sensing cable with coating abrasion.

**Figure 11 sensors-20-06665-f011:**
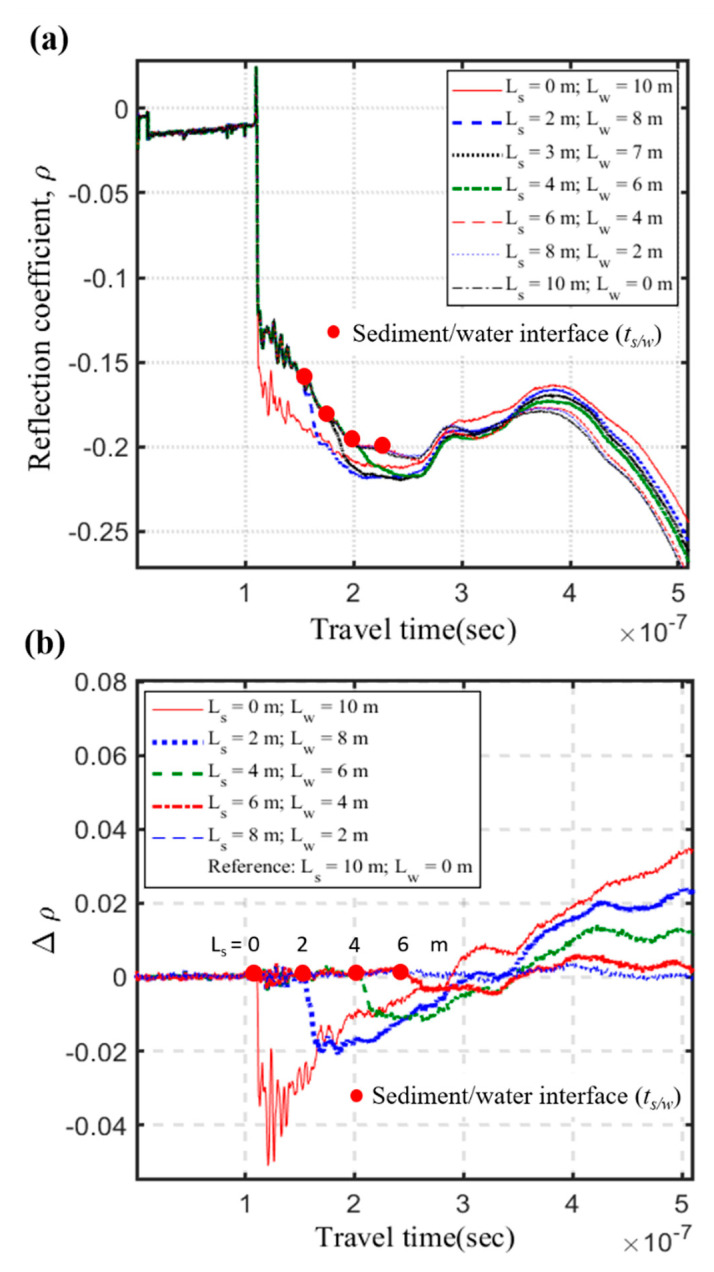
(**a**) Measured raw data of the full-scale scour simulation experiments for the intact TDR sensing cable, and (**b**) the differential waveforms using the case of *L_s_* = 10 m as the reference waveform.

**Figure 12 sensors-20-06665-f012:**
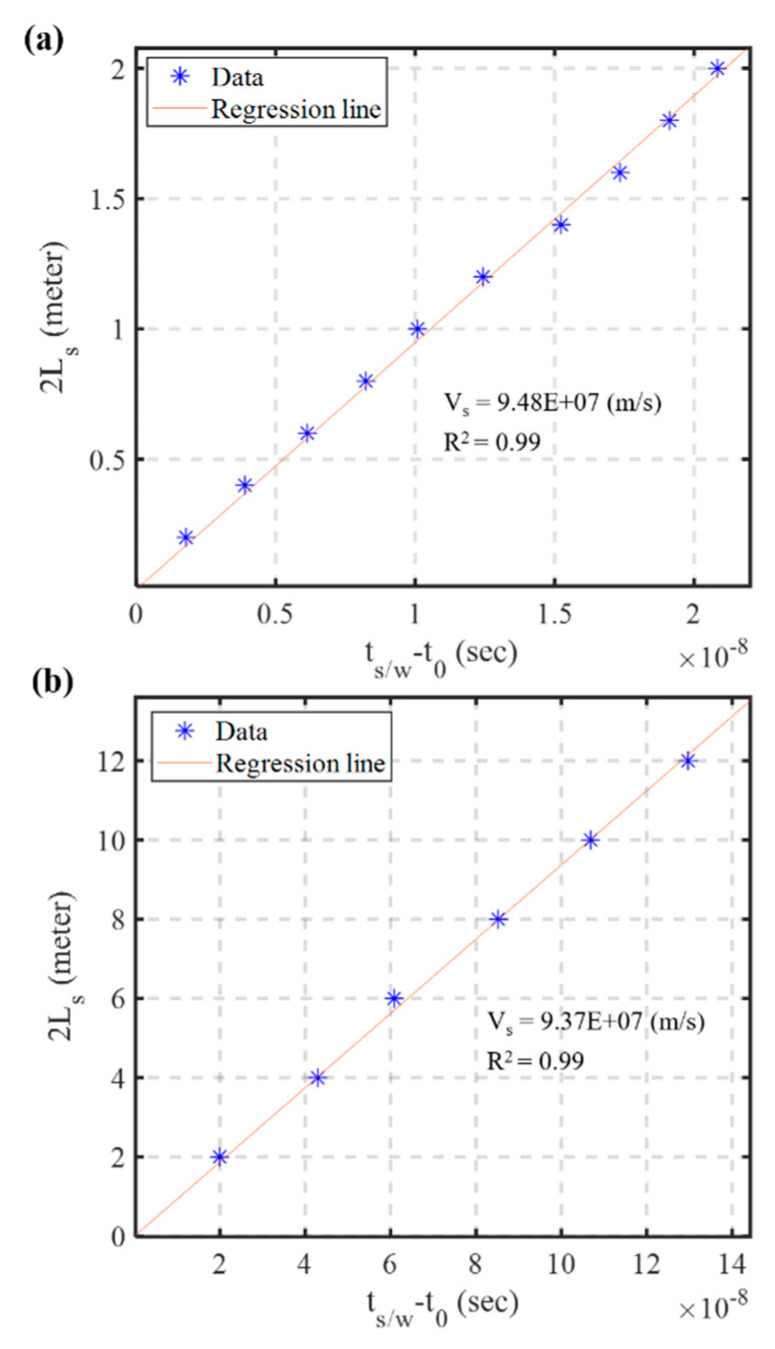
The relationship between 2*L_s_* and round-trip travel time (*t_s/w_* − *t_0_*) in sediments from: the small-scale experiments (**a**); and the full-scale experiments (**b**).

**Figure 13 sensors-20-06665-f013:**
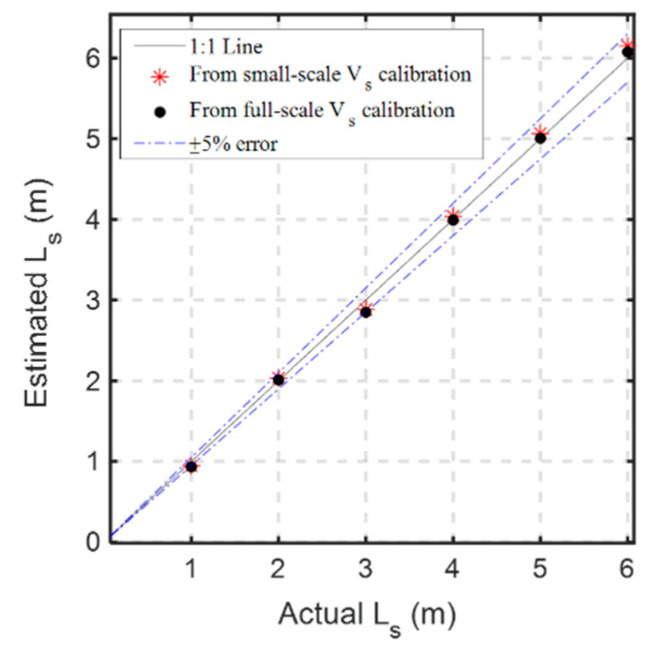
Estimated *L_s_* in the full-scale experiments based on different calibrated vs. in Figure 12.

**Figure 14 sensors-20-06665-f014:**
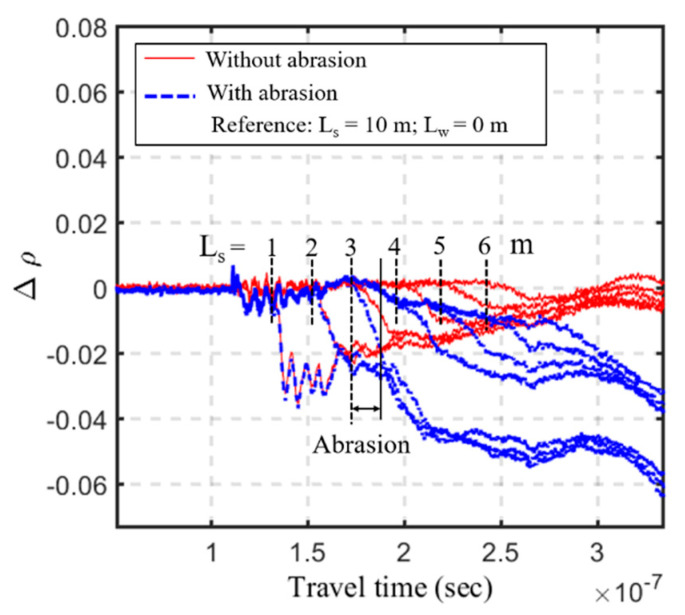
The differential waveforms of the abraded and intact sensing cables at different *L_s_* conditions in the full-scale experiment, taking the same initial waveform (intact sensing cable at *L_s_* = 10 m) as the reference.

**Figure 15 sensors-20-06665-f015:**
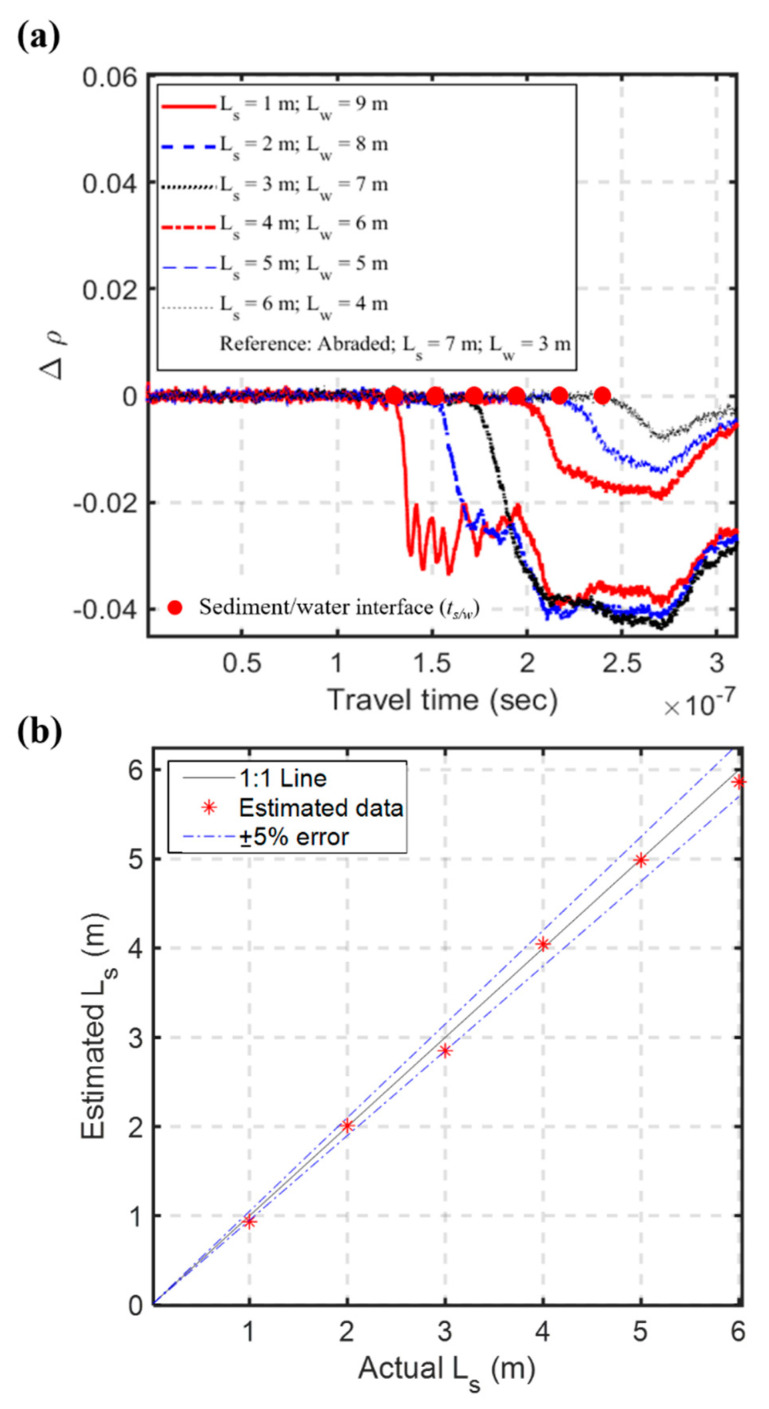
(**a**) Time-lapse differential waveforms after updating the reference waveform with the one after the coating abrasion and sediments refilled to *L_s_* = 7.0 m; and (**b**) the estimated Ls considering recalibration of vs. for the refilled sediments.

**Table 1 sensors-20-06665-t001:** Comparison of scour sensing techniques. (modified from [3,7,8,9,10]).

Device Type	Effect of Environmental Conditions			Continuous Real-Time Monitoring	Deposition Process	Durability
	Temperature	Salinity	Turbidity			
Mechanical sounding rod	No	No	No	No	Yes	Medium
Dropping weight	No	No	No	No	Yes	High
Magnetic Sliding Collar (MSC)	No	No	No	Yes	No	Low
Electrode device	Require remedy	High	High	Yes	Yes	Low
Fiber Bragg Grating (FBG)	High	No	No	Yes	Yes	Medium
Vibration-based turbulent pressure sensors (VTPs)	No	No	No	Yes	Yes	Medium
Dissolved oxygen probes (DO)	High	Unknown	Minor	Yes	Yes	Unknown
Piezoelectric film	No	No	No	Yes	Yes	Medium
Temperature based heat pulse method	High	No	High	No	Yes	Unknown
Amplitude Domain Reflectometry (ADR)	Require remedy	High	Minor	Yes	Yes	Unknown
Time Domain Reflectometry (TDR)	Require remedy	High	Minor	Yes	Yes	Medium
Acoustic Doppler Current Profiler (ADCP)	Minor	Minor	High	No	Yes	High
Sonar	Require remedy	Require remedy	High	Yes	Yes	Medium
Ground Penetrating Radar (GPR)	Minor	High	High	No	Yes	High
Non-intrusive structural vibration monitoring	No	No	No	Yes	Yes	High
Numbered brick	No	No	No	No	No	High
Float out device	No	No	No	Yes	No	High
Smart rocks	No	No	No	Yes	No	High

**Table 2 sensors-20-06665-t002:** The measurement performance evaluation of TDR sensing cables.

Waveguide Type	SNR	Travel Time Per 1.0 m Sand (10^−8^ s)	Apparent *K_a_*
Balanced design			
Original type	10.69	1.661	6.21
Type I	19.04	2.012	9.11
Unbalanced design			
Type II	26.75	2.027	9.24
Type III ^§^	32.37	2.155	10.45

^§^ Chosen for prototyping.

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
