# Peer review of "A New TDR-Based Sensing Cable for Improving Performance of Bridge Scour Monitoring"

_sensors, 2020, doi:10.3390/s20226665_

Round 1
Reviewer 1 Report
The paper presents the application of the refractometry method (TDR) to monitor the condition of bridges. The authors used three different cable (probe) configurations: type I, type II and type III. This study evaluated three new conductor and insulator configurations for constructing the sensing waveguide, including a balanced two-conductor waveguide (Type I), an unbalanced three-conductor waveguide with insulation coating on the middle conductor (Type II), and an unbalanced three- conductor with insulation coating on the two outer conductors (Type III). In all cases, the spacing between the two sets of steel strands was especially enlarged by replacing some steel strands with non-conductor wires to increases measurement sensitivity and avoid shorted condition due to insulation abrasion. The results of the research showed that type III had
best performance in all respects. Type III Caleb has been field tested on larger scale models. The authors proposed a practical application specifying the limits of the detection range and mitigation of the adverse effects of abrasion of the insulation.
The article is interesting from a scientific and cognitive point of view. Below are some comments, the inclusion of which will improve the quality of the article:
1. Chapter 2. Fig.1 c, d - suggest mirror image.
2. Chapter 2 - the authors presented the basics of the TDR method. I suggest supplementing the chapter with information on the influence of variable temperature on the obtained test results.
3. Chapter 3.1 - do the presented results refer to single measurements only or are they average measurements?
4. Chapter 3.1 - with what accuracy was the measurement of the wave transit time?
5. Chapter 3.1 - the concept of Ka coefficient requires a better explanation.
6. Chapter 3.3 - I think that the comparative measurement results require some generalization, allowing for error-free reading and interpretation of measurement results in the future.
7. Chapter 4.2 - the results obtained on large-scale elements almost perfectly match the results of tests of small elements. Can the variation of results be greater in real objects, where the cable lengths will be incomparably greater?
8. Chapter 5 - I believe that some automation in the interpretation of measurement results is necessary in the future.
Reviewer 2 Report
The paper presents a sensor for bridge scour monitoring based on time domain reflectometry. The authors have tested three different types of cables for the sensor.
The paper should highlight the advantages of the proposed sensor in comparison to other commercial existing sensors, preferable to commercial sensors based on time domain reflectometry. This could be included in the section 5 “Conclusion“
Reviewer 3 Report
The article describes the principle of improving the characteristics of the TDR device for bridge scour monitoring by selecting the sensing cable. The final result is the recommendation for specific cable design.
The authors did a fine job of reviewing the literature, reviewing experimental methods, and presenting a general and well-known theory for TDR measurements.
The authors consider three types (designs) of cable versus control cable. The authors carried out experimental work to analyze the quality of the sensitive element in terms of measurement parameters: sensitivity, signal-to-noise ratio, and the possibility of reducing the abrasive effect on the cable.
In my opinion, the overall work is well done, but it does not carry scientific news. No new measurement methods or data processing algorithms have been proposed. The general useful information is in a cable design choosing. The accuracy of determining the area of ​​scouring of the bridge abutment no more than 5% by choosing a cable can be useful also.
The total scientific value of the work lies in solving the engineering task of the optimal design for the sensitive cable selection. It is engineering recommendations only.
I can not recommend this work for publication in the 1Q journal.
Reviewer 4 Report
The paper has a merit for publication. The following issues should be addressed:
1) remove all typos and improve grammar style, see for example line 268 CONDUCOTRS or you mean conductors?
2) include as indicators of performance pSNR and a robust study concerning uncertainty in qualitative (type of uncertainty according to VIM/GUM, guide of uncertainty measurement) and quantity (analytical).
Round 2
Reviewer 2 Report
Comments to the Authors
The idea to use TDR is not new, and there are several references about.
In the previous version of your paper the advantages of your system in comparison to others TDR systems was not highlighted.
According to your replay, you have included at the beginning and at the end of the conclusions.
I understand than you mean lines 464-470 and lines 500-502.
As far as I understand the difference is that the use of TDR allow to monitor the dynamics of the process. But I still don’t know the difference between your proposal and other TDR sensors. Please could you be more precise in the comparison to others TDR existing sensors.
Author Response
The manuscript has been further revised to more precisely point out the different between our proposal and exiting TDR sensors. Please see Line 93-100 in the Introduction.
"The idea of using a TDR-based sensor has been proposed for more than 20 years. However, the usefulness of such techniques and many others has been limited, especially in harsh flow conditions where scour is most critical. Existing TDR scour sensors are rod type, which is unfavorable in high-velocity flow and has very small measurement range. The concept of bundled TDR sensing cable is a big step of change. However, the sensing performance was greatly sacrificed in pursuit of durability and workability. Various aspects of waveguide design should be investigated and experimentally evaluated to improve the sensing performance of a bundled TDR scour cable and confront the adverse effect of coating abrasion and limitation of the measurement range."
Reviewer 3 Report
Dear authors, no doubt, you have made a big work. Your common results and conclusions can present a great interest for engineers. Moreover, the cable selection routine has a basic interest in DTS measurement systems.
However, it should be noted that the 1Q journal must publish only those articles, that contain new methods, new approaches, new algorithms, or new theories.
But with all my respect to the authors and their article, I could not find any scientific novelty in the manuscript. Only two serious conclusions were made by the authors: "A new improved TDR sensing cable was hence proposed based on Type III configuration" and "The coating abrasion does not affect the monitoring of maximum scour depth". These findings do not contain scientific results, which could form a basis for the next researches, in my opinion. The manuscript results do not contain new methods, new approaches, new algorithms, or new theories.
Though the manuscript does not have serious flaws, all experiments were done correctly, all research conducted correctly, I can not recommend publishing this manuscript in 1Q journal.
This manuscript would be excellent for publication in another journal.
Author Response
Dear reviewer, thanks for acknowledging that we have made a big and solid work. The remaining concern is whether this paper scientifically meets the standard of the journal. I have reviewed over 100 papers myself. With all due respect, I am not sure whether it is suitable to draw a line between 1Q and 2Q (or 3Q and 4Q) journals. I would think it’s more important to make sure the paper is “original”, meets the “scope” of the journal, and more importantly presents a “breakthrough”. Furthermore, besides methods and theories, the importance of experimental evaluation shouldn’t be undervalued either.
The definition of novelty is also arguable. Yes, the TDR method for scour monitoring is not new. In fact, the idea of using a TDR-based sensor has been proposed for more than 20 years. Yet, it has not made any significant impact to the bridge scour monitoring. Existing TDR scour sensors (probes) are rod type. They are unfavorable in high-velocity flow and have a very small measurement range, not to mention practical problems of installation and debris. The recent concept of bundled TDR sensing cable is a big step of change. However, the sensing performance was greatly sacrificed in pursuit of durability and workability. This study proposed a NEW TDR scour sensor through various waveguide considerations and thorough experimental evaluation. It presents a NEW “SENSOR” that has not been seen before. We believe this is a “breakthrough” to the TDR-based method for scour monitoring. With the new sensor, the measurement sensitivity is much improved and the effect of insulation abrasion is minimized while taking advantage of the anchored sensing cable concept. The experimental evaluation of sensing range and coating abrasion, which was never addressed in existing TDR probes, is also valuable for actual TDR sensor deployment. To this end, we confidently believe this paper is “original”, meets the “scope” of the journal, and presents a “breakthrough”.
The title of our paper has been changed to “A New TDR-based Sensing Cable for Improving Performance of Bridge Scour Monitoring” to better describe this study’s contribution. The Introduction and Conclusion were also revised to better highlight the difference between our proposal and existing TDR sensors. Please see Line 93-101 and Line 465-472.